# Collateral Damage in the Placenta during Viral Infection in Pregnancy: A Possible Mechanism for Vertical Transmission and an Adverse Pregnancy Outcome

**DOI:** 10.3390/diseases12030059

**Published:** 2024-03-20

**Authors:** Victor Javier Cruz-Holguín, Luis Didier González-García, Manuel Adrián Velázquez-Cervantes, Haruki Arévalo-Romero, Luis Adrián De Jesús-González, Addy Cecilia Helguera-Repetto, Guadalupe León-Reyes, Ma. Isabel Salazar, Leticia Cedillo-Barrón, Moisés León-Juárez

**Affiliations:** 1Laboratorio de Virologia Perinatal y Diseño Molecular de Antigenos y Biomarcadores, Departamento de Inmunobioquimica, Instituto Nacional de Perinatología, Mexico City 11000, Mexico; victor.cruz@cinvestav.mx (V.J.C.-H.); biol.didier@gmail.com (L.D.G.-G.); manugenes18@hotmail.com (M.A.V.-C.); 2Departamento de Biomedicina Molecular, Centro de Investigación y Estudios Avanzados del IPN (CINVESTAV), Mexico City 07360, Mexico; lcedillo@cinvestav.mx; 3Posgrado de Inmunología, Escuela Nacional de Ciencias Biologócas (ENCB), Instituto Politecnico Naciona, Mexico City 11350, Mexico; misalazar@ipn.mx; 4Laboratorio de Inmunologia y Microbiologia Molecular, Division Academica Multidisciplinaria de Jalpa de Méndez, Jalpa de Mendez 86205, Mexico; haruki.arevalo@ujat.com; 5Unidad de Investigación Biomédica de Zacatecas, Instituto Mexicano del Seguro Social, Zacatecas 98000, Mexico; adrian_6101@hotmail.com; 6Departamento de Inmunobioquimica, Instituto Nacional de Perinatología, Mexico City 11000, Mexico; addy.helguera@inper.gob.mx; 7Laboratorio de Nutrigenética y Nutrigenómica, Instituto Nacional de Medicina Genómica (INMEGEN), Mexico City 14610, Mexico; greyes@inmegen.gob.mx; 8Laboratorio Nacional de Vacunología y Virus Tropicales (LNVyVT), Escuela Nacional de Ciencias Biologócas (ENCB), Instituto Politecnico Naciona, Mexico City 11350, Mexico

**Keywords:** virus, placenta, trophoblast, pregnancy, maternal–fetal interface

## Abstract

In mammals, the placenta is a connection between a mother and a new developing organism. This tissue has a protective function against some microorganisms, transports nutrients, and exchanges gases and excretory substances between the mother and the fetus. Placental tissue is mainly composed of chorionic villi functional units called trophoblasts (cytotrophoblasts, the syncytiotrophoblast, and extravillous trophoblasts). However, some viruses have developed mechanisms that help them invade the placenta, causing various conditions such as necrosis, poor perfusion, and membrane rupture which, in turn, can impact the development of the fetus and put the mother’s health at risk. In this study, we collected the most relevant information about viral infection during pregnancy which can affect both the mother and the fetus, leading to an increase in the probability of vertical transmission. Knowing these mechanisms could be relevant for new research in the maternal–fetal context and may provide options for new therapeutic targets and biomarkers in fetal prognosis.

## 1. Introduction

When a new village emerged in Celtic culture, they used to plant trees in the town center which would represent protection and prosperity. During human gestation, the placenta plays a role similar to that of a highly specialized organ, creating an environment suitable for fetal growth and development. The placenta is a hybrid organ comprising fetal and maternal tissues [1]. It has three main functions: (a) maintaining homeostasis between nutrients and waste in the fetus, (b) regulating the fetal environment and maternal immune response to avoid rejection, and (c) establishing a barrier against microorganisms [2,3]. To fulfill this purpose, cells constituting the placenta arrange themselves to create anatomical and functional units defined as chorionic villi. Two structures organize inside these villi: floating villi, which are in intimate contact with maternal blood, regulating the transport of nutrients, gases, and waste between the fetus and the mother, and anchoring villi, which form a physical link with the uterine wall, providing structural support between the mother and the fetus [4].

The principal cellular component of the placenta is the trophoblast, which is a progenitor of other trophoblast types that participate in the development of chorionic villi throughout pregnancy. In this respect, the trophoblast stem gives rise to cytotrophoblasts (CTBs), which are proliferative mononuclear cells that leave the basement membrane of the villi and thus generate ramifications of anchoring and floating villi. In addition, CTBs are progenitors of the syncytiotrophoblast (STB), a multinuclear, contiguous cell layer that covers the entire surface of the floating villi. Interestingly, during pregnancy, the placenta undergoes several morphological changes that generate the villous architecture. At the end of the first trimester, the transition to a hemochorial placenta, in which maternal blood has intimate contact with the fetus, requires a specialized process carried out by another type of trophoblast, the extravillous trophoblast (EVT) [1,5]. These cells are distributed in anchoring villi, organize into nonpolarized mononuclear columns, attach to the uterine wall to remodel the maternal microvasculature, and express L-selectin and carbohydrates that maintain column integrity and facilitate cell motility. EVTs can migrate and replace endothelial cells in the uterine veins and arteries to create wide-bore, low-resistance blood vessels that divert blood flow from the mother to the surface of the placenta (Figure 1) [6,7].

The cellular organization of the placenta represents an almost unbreakable barrier to pathogens by delimiting the maternal and fetal compartments and the development of immunological mechanisms that participate in the surveillance and control of microorganisms [8,9,10]. However, some pathogens, such as viruses, have developed strategies to penetrate this biological barrier, allowing for vertical transmission to the fetus. Viruses such as Zika (ZIKV), Human Cytomegalovirus (HCMV), Rubella, Human Immunodeficiency Virus (HIV-1), Herpes Simplex Virus 1 and 2 (HSV-1,2), and Human Parvovirus B19 have been widely documented as pathogens that may have a broad tropism for cellular components of the placenta [11]. Infected placental cells enhance viral load in this region, leading to fetal transmission. In addition, they are related to congenital infections, which are associated with a disruption in fetal organogenesis and often result in the death of the product. However, the possibility of vertical infection with some viruses remains controversial. 

An apparent effect on the placenta was observed through the generation of macroscopic and microscopic morphological changes at this interface, including the presence of fibrin deposits, villitis and intervilliitis, chorioamnionitis, malperfusion placental, chronic histiocytic intervillositis, and other illnesses [12,13,14]. This pathological manifestation can functionally and structurally disrupt the physiological barrier, culminating in complications ranging from abortion, premature birth, fetal growth restriction, the premature rupture of membranes and fetal and maternal death. In this review, we aimed to capture evidence related to the effects of some viruses on the generation of pathological disturbances in the placenta, following the idea that not all viral infections can be transmitted vertically but maternal infection can collaterally damage the architecture of the placental barrier, culminating in adverse perinatal outcomes [15,16].

## 2. The Temporality of Pregnancy and the Effect of Viral Infection on the Placenta

Clinical evidence has shown that the time of gestation at which a woman acquires a viral infection directly correlates with the effect on and susceptibility of the placenta (Figure 2). Additionally, multiple factors will participate so that the virus directly affects the fetus since viral tropism, the stage of fetal development, and the immunological regulation that occurs in pregnancy will actively participate in this phenomenon [17]. In this sense, the access route for viruses that promote vertical infection and reach fetal circulation has been mainly characterized by a maternal hematogenous pathway. The maternal blood in contact with the villous tree through the floating villi or the maternal decidua interacts with the anchoring villi, culminating in the placental stroma, where the fetal blood capillaries represent the last frontier for viruses to be transported toward fetal circulation [18,19]. Viruses can also be internalized at this interface by the amniotic sac through the decidua parietalis or the maternal cervix [20]. However, as previously mentioned, the placental architecture changes throughout gestation. Maternal blood is not in contact with the chorionic villus until the second trimester of pregnancy. Therefore, it is likely that viral access during the first trimester involves the anchoring villus or decidua parietalis.

In this context, crucial development of the placenta and organogenesis occur during the first trimester of pregnancy (0–13 weeks). Therefore, there is a greater probability that the fetus will be affected or develop considerable damage from an infection. Various viruses are associated with damage to placental development during this stage, which is associated with poor implantation and spontaneous pregnancy loss. In the case of HCMV, a direct association has been observed with pregnancy loss in the first trimester [21,22]. Infection directly affects EVTs, which are related to the implantation of villi anchored to the endometrium. HCMV infection in these trophoblasts modifies their proliferative, migratory, and invasive properties necessary for implantation [23]. This change is associated with a mechanism involving the transforming growth factor-β (TGF-β)/Smad pathway, which controls placental maturation in the maternal endometrium [24]. 

Immune tolerance at the maternal–fetal interface is crucial for achieving a successful pregnancy. The presence of paternal antigens, which maternal cells can recognize as foreign factors and activate an immune response against, is regulated by the expression of human leukocyte antigen G (HLA-G). This molecule regulates the functions of immune cells, leading to communication that prevents damage to the fetus. Gestational loss in the first trimester of pregnancy is associated with a defect in HLA-G function and a lack of regulation of immune tolerance [25]. The effects of certain infections on this process have also been studied. Evidence regarding HSV has shown that infection in trophoblasts disturbs the transport of antigens via HLA-G, which could affect the tolerance of immune cells and thereby generate mechanisms of damage to the maternal–fetal interface, culminating in infection-associated abortions [26,27,28]. 

Another phenomenon associated with early gestational loss caused by a viral infection in the first weeks of pregnancy or the process of conception is human papillomavirus (HPV). A correlation between the incidence of abortion and HPV infection has been demonstrated [29,30,31]. Animal studies have also shown the possibility of fertilization with sperm carrying a plasmid containing the HPV genome, which was identified in the embryo after fertilization [32,33]. Additionally, other evidence suggests that HPV infection promotes apoptosis in the embryos of mice infected with the virus [34]. Evidence from placentas obtained from abortions of HPV-infected women has shown trophoblast infection and an impact on migration and apoptosis in cellular models of HPV-infected trophoblasts. Therefore, this virus may be associated with fertilization, implantation, and early embryonic development.

A crucial event in pregnancy is the dynamic change in the composition of immune cells and the expression of cytokines at the maternal–fetal interface which is related to the temporality of gestation. In the first trimester, implantation and the development of the placenta require an increase in inflammatory mediators and an accumulation of immune cells in the maternal decidua. Later, in the second trimester (14–26 weeks), an anti-inflammatory profile is required at the maternal–fetal interface which is associated with a change in the recruitment of macrophages with an M2 phenotype, decidual NK cells, and regulatory T cells that participate in maintaining immunological tolerance to the fetus [35,36,37]. Viral infections, however, can significantly perturb this fine regulation by mounting an antiviral response at the maternal–fetal interface which promotes placental damage or fetal development [14]. 

Research related to acute viral infections in pregnant women has shown that the immune evasion mechanisms evolved by some viruses are crucial for participating in a switch in the context of maternal immunological regulation, thus significantly affecting pregnancy. Lassa virus infection during pregnancy has been strongly associated with maternal death and pregnancy loss in the third trimester of pregnancy [38,39,40]. Studies focusing on understanding how this virus regulates the innate immune response are crucial to identifying its effect on pregnancy. Specifically, it has been determined that the Lassa virus can regulate the interferon-α (INF-α) response by expressing its matrix and nucleocapsid proteins. Negative regulation owing to a lack of interferon response in pregnant women can block the activation of adaptive immunity to counter the Lassa virus, thereby promoting maternal death. However, the regulation of the interferon response during viral infection contributes to a failure in the immunoregulatory role of the maternal–fetal interface [41,42,43].

Similarly, during outbreaks of Ebola infection, higher percentages of maternal mortality were reported and increases in preterm births and abortions were documented [44,45,46]. The Ebola virus also negatively regulates the innate immune response and viral proteins, such as VP34 and V24, which mediate the detection of viral RNA by blocking the activation of STAT pathways, culminating in a failure of the antiviral response to interferons [47,48]. As mentioned above, pregnancy requires optimum immunological regulation that can be mediated by cellular components of the fetal–maternal interface and microbiota which contribute to establishing immunological balance, protecting the fetus against damage and inducing specific tolerance to paternal antigens. An interesting example of this regulation and how viral infections can affect this immune balance is associated with the effect of LPS on the microbiota of the maternal–fetal interface and the production of INF-β-induced activation via TLR-4, which could mediate the control or stimulation of apoptosis in the trophoblast during normal pregnancy (Figure 2) (Table 1) [8,49].

## 3. Histopathological Changes in the Placenta Caused by Viral Infections

Viral infections during pregnancy trigger a series of histological alterations that affect placental homeostasis. Among the most notable alterations are fibrin deposits, placental calcification, fibrosis, hemorrhage, intervillositis, chorioamnionitis, and inflammation due to the infiltration of immune cells (Figure 3) (Table 2).

Fibrin deposits: The placenta is a highly vascularized organ which is subject to structural modifications during pregnancy. When this tissue suffers damage, coagulation can occur. In some instances, if this process is exacerbated, fibrin derived from fibrinogen leads to thrombosis, necrosis, and placental infarcts [81,82,83]. It has also been reported that fibrin deposition between the chorionic villi can promote apoptosis in the syncytiotrophoblast layer, structurally discontinuing this protective barrier [84].

In this sense, infection by some viruses can induce the release of proinflammatory molecules that result in the activation of the coagulation cascade, thus causing thrombosis derived from inflammation or thromboinflammation [50]. For example, during infection with some viruses such as Adenovirus Coxsackievirus-B and severe acute respiratory syndrome coronavirus 2 (SARS-CoV-2), protease-activated receptors (PARs) such as PAR1 and PAR2 can regulate coagulation as an immune response to the infections since PAR1 positively regulates the expression of protein prothrombotic factors such as VWF and P-selectin and platelet factor 4 of endothelial cells [51,52] and the expression of tissue factor (TF) via the detection of the presence of the intracellular viral genome [66].

Placental calcification: In recent years, interest in calcium crystal deposits has been increasing; while the pathology of this phenomenon is not yet completely understood, it has been reported to be associated with placental dysfunction and poor pregnancy development [85].

The normal process of calcification is mainly controlled by molecules called bone morphogenic proteins (BPMs) which play an important role in fetal development, such as the molecule called BMP7. BMP7 is expressed by cytotrophoblasts in early stages due to its implantation in the uterus [86,87] but also by other proteins such as prostate-derived factor (PDF), placental BMP (PLAB), and insulin-like growth factor (INSL-4). In addition to being involved in embryogenesis, these molecules can also promote the formation of bone and cartilage, and their expression is found basally in the placenta during gestation [88,89]. 

Dystrophic calcification mainly occurs due to tissue undergoing a process of necrosis or apoptosis. When caused by necrosis, the damaged or dysfunctional cell membranes allow for the passage of calcium from the intracellular section to the extracellular section and its combination with phosphate, thus producing hydroxyapatite crystals that accumulate in the tissue [90,91]. In the case of apoptosis, calcium is concentrated in apoptotic bodies that appear as deposits of calcium crystals in formation [92,93]. The formation of these calcium deposits can damage blood flow, leading to poor chorionic vasculature. However, these deposits are also associated with placental infarctions, uterine growth restriction, postpartum hemorrhage, low birth weight, placental abruption, chorioamnionitis, and neonatal death [85,94,95]. In viral infections during pregnancy, this phenomenon can induce the formation of calcium deposits owing to the ability of these viruses to induce apoptosis or necrosis in their target cells, mainly trophoblasts.

Placental inflammation: During viral infections, target cells can activate an immune response, thus promoting the infiltration of cells such as leukocytes into the tissue. When this phenomenon occurs for a prolonged period, it can induce alterations in the architecture of the placenta, culminating in pathological states, such as intervillositis or chorioamnionitis, depending on the tropism of the virus to the different cellular populations of the placenta.

Intervillositis is a rare and poorly understood condition characterized by the infiltration of immune system cells, such as monocytes and lymphocytes of maternal origin, into the chorionic villi, thus generating a localized inflammatory environment which is associated with the restriction of intrauterine growth and fetal death [83,96].

Chorioamnionitis: This is an acute inflammatory condition in the membrane of the chorion, amnion, or both caused by the infiltration of immune system cells, which generates an inflammatory environment. This condition can promote membrane rupture and modify placental homeostasis, thus causing adverse effects such as fetal death and premature birth [97,98].

Syncytial knots: This condition is characterized by the focal aggregation of syncytial nuclei on the outer surface of chorionic villi whose formation process is closely related to cell death processes such as apoptosis and necrosis through oxidative damage mechanisms [99,100,101]. It has also been associated with the development of preeclampsia, preterm birth, poor fetal nutrition, and severe infection by viruses such as SARS-CoV-2 [53,73,102].

Maternal vascular malperfusion (MVM) of the placenta is damage that impairs the intervillous blood supply. MVM is characterized histologically by damage to the decidual blood vessels, causing abnormal remodeling in the coiling of the maternal spiral artery as well as in the villous parenchyma, leading to altered oxygenation and blood flow dynamics in the intervillous space. This phenomenon can cause fetal oxygenation and malnutrition, which directly affect fetal development during pregnancy. This malformation of the maternal arterial architecture can also induce the development of vascular occlusions that lead to hypoxic–ischemic injury and heart attacks [103]. The etiology of this condition is not yet completely understood; however, it has been associated with inflammatory processes such as viral infections [74,103,104].

Placental abruption refers to the premature separation of the placenta from the myometrium, associated with the premature rupture of membranes and both maternal and fetal morbidity and mortality, promoting blood loss and intravascular coagulopathy [105,106]. In some cases, meta-analyses have reported that viral infections such as SARS-CoV-2 and hepatitis B virus (HBV) are associated with a high risk of placental abruption [77,78].

Hemorrhage: Tissue bleeding is mainly promoted by alterations in the vascular permeability of the endothelium, which can be due to various etiologies. In the viral context, viruses such as Filovirus, Flavivirus, Arenavirus, and Bunyavirus can induce hemorrhage through various mechanisms. Some viruses can induce the expression of adhesion molecules such as ICAM-1 or VCAM-1 in endothelial cells, which causes an increase in the permeability of the endothelium. Furthermore, inflammatory conditions with the presence of cytokines such as TNF-α, IL-8, and IL-6 cause greater permeability, consequently causing tissue hemorrhage [107,108].

## 4. Viral Infections That Cannot Be Transmitted Vertically during Pregnancy

According to clinical evidence, some viral infections during pregnancy can induce complications such as premature birth and membrane ruptures and even the death of the mother. In this section, we compiled reported data on viruses that do not usually have the capacity to be transmitted vertically; however, when they infect humans during pregnancy, they can cause damage, mainly to placental tissue, which affects the correct outcome of the pregnancy (Figure 3) (Table 2).

### 4.1. Hepatitis E Virus (HEV)

HEV, classified as *Orthohepevirus A*, causes acute and chronic inflammation of the liver, which has been identified as a public health concern worldwide. However, pregnant women are susceptible to serious clinical outcomes during infection with this virus, mainly due to genotypes 1 and 2, which have been reported worldwide to have a 25% mortality rate in mothers, unlike other types of causative hepatitis viruses. Some cases have been reported: nineteen pregnant women were infected with HEV, and seven newborns which presented icteric, anicteric, and with hyperbilirubinemia and premature birth died in the first week. Of all newborns, only nine survived [54]. Infections with this virus during pregnancy have been reported to be 20–30%. This high mortality rate has been mainly associated with the production of inflammatory cytokines in the placental tissue, causing fetal death or spontaneous abortion [55,56,57,109]. Damages caused by infection with this virus in the placentas of pregnant women mainly include calcification, fibrosis, tissue inflammation, membrane ruptures, and hemorrhages [57].

### 4.2. Dengue Virus (DENV)

DENV belongs to the *Flavivirus* family and is mainly transmitted by mosquito vectors of the Aedes genus. This agent can cause hemorrhagic fever in severe cases. To date, clinical cases of infection with DENV have been reported during pregnancy but are not considered capable of being transmitted vertically. Only a few known cases are considered isolated in vertical transmission, and it has been reported that the risk factor in pregnant women exposed to the virus has a risk ratio of 3.4 (RR) for neonatal death and 6.8 (RR) for early neonatal death, with pregnant women exposed to DENV having an increased risk of maternal death, neonatal death, and infant death after birth [58]. Other studies have reported that approximately 9% of pregnant women infected with this virus experience fetal death, while 4.5% experience postnatal death [110]. However, there are reports of pregnant women infected with DENV in which placental histological analyses have shown signs of hypoxia, choriodeciduitis, deciduitis, and intervillositis, while the viral antigen was found in trophoblasts [111]; infiltration of cells of the immune system, such as macrophages and TCD8+ lymphocytes, causing severe damage to the tissue due to exacerbated inflammation, necrosis, hemorrhage, and edema, was also found [59]. 

### 4.3. Chikungunya Virus (CHIKV)

CHIKV belongs to the alpha virus family and is transmitted mainly by mosquito vectors of the Aedes genus. Clinically, this virus can cause high fever and joint pain. To date, few cases of CHIKV infection during pregnancy have been reported. However, in recent years, the effects of infection with this virus have been described in a maternal–fetal context. It is generally known that when this virus infects a pregnant woman, it is the causal agent in 4.75% of miscarriage and stillbirth cases, [112] and other clinical aversions; for example, cases of severe dermatological manifestations and low body weight have been reported during the third trimester of pregnancy. Furthermore, histopathological aversions have described that the antigen and genetic material from the virus were found in the placental tissue, causing deciduitis, a delay in maturation and differentiation of the chorionic villi with an absence of the syncytiotrophoblast layer, interruptions in the placental surface, fibrosis in the stromal region, collapsed vessels of the stem villi, calcification of and damage to the stem villi, thickening in the basal layer of the endothelium, mitochondrial and endoplasmic reticulum trophoblast alterations trophoblasts, subtrophoblastic edema, and cell death [79]. All these aversions are considered unsuitable for promoting the exchange of gases and nutrients between the mother and fetus [67,113]. Despite this, it has been reported in some meta-analyses that approximately 50% of infected pregnant women can vertically transmit the virus; most of these cases involve intrapartum transmission at the time of delivery (75 cases by cesarean section and 96 cases by natural delivery). In the majority of these cases, there was little or no presence of the viral antigen in the placenta but high viremia, and transmission occurred through the blood supply [112].

### 4.4. Respiratory Syncytial Virus (RSV)

RSV is a respiratory virus belonging to the *Orthopneumovirus* family. It mainly affects children and older adults and is transmitted horizontally through the respiratory tract, with the lungs and upper respiratory tract being the main target organs. However, it can spread to other tissues such as the placenta during pregnancy through maternal blood, and in a few known cases, alterations occur in this tissue. Epidemiological studies have reported that RSV infection in pregnant women occurs in 2–9% of pregnancies [114,115]. Fonceca et al. reported higher levels of RSV expression in human cord blood samples collected in winter than in samples collected during non-winter months [60]. Additionally, RSV-positive pregnant women experience more frequent preterm births than RSV-negative pregnant women [61].

In this context, it has been reported that RSV can infect cytotrophoblasts and is more permissive to fibroblasts and Hofbauer cells, mainly because it uses macrophages as a reservoir in the placental tissue. Histopathology can also promote alterations in the membranes of syncytiotrophoblasts [62,116]. However, data showed a difference in the permissiveness of the virus in placental cells between different donors, suggesting that this phenomenon is due to epigenetic and genetic background and not only immune system activity or RSV virulence [116]. A lack of clinical reports on RSV infection during pregnancy limits the diagnosis of this pathological agent.

### 4.5. Human Bocavirus (HBoV)

This virus belongs to the *Parvoviridae* family, with four distinct subtypes: bocaparvovirus type 1 is classified into HBoV1 and HBoV3, whereas bocaparvovirus type 2 is classified into HBoV2 and HBoV4. HBoV is the second most common parvovirus which is pathogenic to humans and causes gastroenteritis and respiratory tract aversion in adults and infants [117]. Other symptoms include respiratory failure, pneumothorax, myocarditis, and dermatological manifestations [69]. HBoV has an antibody seroprevalence of up to 95% in children (≥5 years) [118] and >94% in adults [119,120]. In adults, HBoV is rarely detected due to its high seroconversion rate [117].

The consequences of HBoV infection in pregnant women and their fetuses are unknown [121]. However, an association of the virus with spontaneous abortions in its natural hosts, with an incidence of approximately 25–45% in the placentas of pregnant women diagnosed positive for the virus, has been reported in recent years [63].

### 4.6. Ebola Virus (EBOV) and Other Hemorrhagic Fever Viruses

The Ebola virus belongs to the family Filoviridae [122]. This deadly virus is transmissible through direct contact with blood and other fluids (saliva, sweat, breast milk, and semen) [123]. The main symptoms of infection are high fever, muscle pain, vomiting, diarrhea, skin rashes, and internal and external bleeding, which culminate in death in most cases. In pregnancy, cases positive for EBOV are associated with maternal hemorrhage, preterm labor, miscarriage, stillbirth, neonatal death, and high mortality rates [124]. This virus is capable of infecting the placenta during pregnancy, with a 100% mortality rate observed in fetuses [64,70], and has been found in the layer of cytotrophoblasts and syncytiotrophoblasts. Infiltration of immune system cells such as macrophages in the intervillous space has also been reported, with malformations of the nuclei in trophoblasts [65,125]. 

For viruses that cause hemorrhagic fever, such as the Marburg, Lassa, and Rift Valley Fever viruses (MARV, LASV, and RVFV, respectively), it has been reported that infection in human pregnancy has a high probability of stillbirth. Because placental cells express the viral receptors necessary for VHF infection to be productive, it has been associated with a high rate of fetal death during pregnancy [126]. MARV, a related filovirus, has few reports of infection during pregnancy; however, the majority of pregnant women with this virus do not survive [70,124]. LASv is a virus that belongs to the Arenaviridae family; it is able of being transmitted zoonotically or from person to person through body fluids [127]. Infection with this virus during pregnancy has been associated with spontaneous abortions, with an incidence rate of up to 95% [80]. With respect to RVFV, epidemiological data on the rates of transmission among pregnant women infected with this virus are limited. Some studies found little association between RVFV infection and pregnancy. In Mozambique in 1981, the rates of RVFV-specific antibody prevalence were similar between women who experienced and those who did not experience late-term spontaneous fetal loss. However, interestingly, 24% (5/21) of seropositive women were those who had a fetal loss compared to 15% (143/969) of seronegative women [128]. Other relevant studies showed that a pregnant woman whose serological test using peripheral and umbilical cord blood confirmed an RVFV infection had main symptoms characteristic of an infectious fever [129]. On the other hand, studies have reported that RVFV infection during pregnancy is four times more likely to present miscarriage compared to CHIKV infections and was also associated with the development of late-term spontaneous abortion or stillbirth [130].

### 4.7. Human Coronavirus

Some of the most studied coronaviruses that have caused major pandemics worldwide have also been capable of causing complications during pregnancy by directly affecting the placental tissue.

The Middle East respiratory syndrome (MERS) coronavirus, for which we have limited knowledge about the perinatal characteristics of infection, it has been documented that approximately 91% of pregnant women presenting clinical complications of infection with this virus can experience fetal or post-birth death [131,132]. Other clinical characteristics of MERS-coronavirus infection include abrupt vaginal bleeding, membrane rupture, and placental abruption [133,134].

For SARS-CoV-1, cases of severe infection in pregnant women have been reported with impacts on placental formation, spontaneous abortion, or maternal death [135,136]. For example, reports have indicated that an increase in the deposition of fibrin in the subchorionic and intervillous spaces can impact blood supply from the mother to the placenta, causing extensive fetal thrombotic vasculopathy with abnormalities in the villus vasculature that result in poor vascular perfusion and high tissue inflammation [137].

Since the 2019 SARS-CoV-2 pandemic, cases of infection with this respiratory virus in pregnant women and its effects on placental and fetal development have been reported. Vertical transmission has been rare, with a rate of 2–6.45%. Approximately 71.4% of placentas in infected pregnant women were reported to test positive for viral material by a reverse-transcription polymerase chain reaction (RT-PCR) [138,139,140]. The presence of genetic material and antigens in placental tissue has been reported, as well as various histological alterations that include poor MVM, increased fibrin deposition, villous agglutination, intervillous thrombi, atherosis, and increased syncytial knots [141,142]. Other studies have shown the presence of extensive calcifications throughout the placenta [143], It has also been reported that infections with this virus during the first trimester can cause placental damage associated with inflammation, manifesting as intervillitis, with high infiltration of neutrophils and Hofbauer cells in the chorionic villi as well as cell death [68,75]. The induction of natural proinflammatory elements during pregnancy, in addition to the response against viral infection, suggests that a cytokine storm in this tissue could complicate placental damage during pregnancy. Moreover, it has been reported that severe SARS-CoV-2 infection in pregnant women can induce an increase in the formation of syncytial knots in the chorionic villi, causing hypoxia in the placental tissue and putting the correct development of pregnancy at risk [73].

Due to the nature of infection, limited studies have been conducted on these viruses in the maternal–fetal context. Therefore, investigating viruses that have negative aversions on the placental tissue without vertical transmission, affecting pregnancy development or even leading to death, is crucial.

## 5. Viral Mechanisms Could Cause Potential Damage to the Placenta

As previously described, the viruses mentioned above can cause histopathological and functional alterations in the placenta during pregnancy. Therefore, in this section, we focus on describing viral mechanisms to explain how these alterations occur in the placental tissue.

### 5.1. HEV

In vitro studies have demonstrated that HEV is capable of infecting human placental trophoblastic cell lines such as JEG-3, cytotrophoblasts, and syncytiotrophoblasts from placental explants. This is believed to be due to a decrease in the presence of type I interferons, which makes the placenta more permissive to viral infection [71,144]. Furthermore, important hormonal changes that occur during pregnancy have immunosuppressive effects. HEV is capable of increasing the expression of estrogen, progesterone, and β-HCG, successfully colonizing the placental tissue [56]. These results demonstrate the tropism of this virus in the placental tissue.

When HEV infects trophoblasts during pregnancy, studies have shown that the virus induces the expression of proinflammatory cytokines such as IL-1β and IL-18 in the serum and placenta of pregnant women and in infected macrophages [71]. In addition to the activation and overexpression of TLR3 (related to cell death), trophoblastic tropism explains why inflammation generates the early rupture of membranes, which is why pregnant women infected with HEV present high levels of infiltration of immune system cells and necrosis in the chorionic villi [71,145,146].

### 5.2. DENV

Because maternal blood should not mix with that of the fetus without first passing through membranes composed of trophoblasts, the integrity of the structure of the chorionic villi is important for restricting infection by pathogens, such as DENV. In this context, DENV begins by preferentially infecting cells of the maternal decidua, which participate in the remodeling of the arteries and placentation. By becoming a target of infection, this virus can cross the placental barrier and reach the chorion [147]. With the ability to infect the trophoblasts in the chorionic villi, DENV can induce the activation of the immune system and promote the release of proinflammatory elements such as TNF-α, RANTES/CCL5, MCP1/CCL2, and VEGF/R2 that can alter vascular permeability, disrupting the architecture of the chorionic villi and promoting the spread of the virus to both the placenta and the fetus itself, causing fatal aversions such as fetal death [59,148].

On the other hand, it is known that under normal conditions, the response par excellence to viral infections in the placenta is mainly mediated by type I interferons (INF-Is) which help activate the anti-viral state and counteract viral infection; however, DENV has the ability to inhibit the INF-I signaling pathway owing to its non-structural proteins NS4A/NS4B, NS2B, and NS5, which inhibit the activation of STAT-1 and 2, thus reducing the anti-viral response [149,150]. It has also been shown that nonstructural proteins, such as NS4A/NS4B and NS2B, can inhibit the activation of the cGAS/STING pathway, which modulates the production of INF-Is [151]. This was demonstrated in cellular systems such as first-trimester extravillous trophoblasts (HTR8) and THP-1 monocytes, which have a regulated interferon response when infected with DENV [150]. The evasion of the immune response, mainly by interferons, is relevant because it could lead to the replication of this virus in the placental tissue, where it is easier to spread.

Other studies conducted in vitro, using the HTR8 trophoblast cell line, showed that DENV infection can decrease the activation of the mTOR pathway, which is related to cellular processes such as growth and differentiation compared to other flaviviruses. This growth and differentiation process is crucial for the formation of chorionic villi throughout gestation and may be associated with histopathological aversions of placental structures if affected, which could be related to spontaneous abortions [152]. 

### 5.3. CHIKV

Viruses such as Chikunguya, which have not been associated with vertical transmission, have been shown to infect placental cells such as trophoblasts and decidual, stromal, Hofbauer, and endothelial cells. This infection promotes an overexpression of proinflammatory elements associated with the development of severe damage in the placental tissue, causing inflammation in the decidua, villous edema, villous necrosis, dystrophic calcification and thrombosis, which culminate in alterations in placental homeostasis and promote histopathological alterations, evidencing a harmful environment for the fetus and the mother [16]. It is believed that the infection of the placenta by this virus occurs through a microtransfusion of maternal blood and the subsequent rupture of the membrane of syncytiotrophoblasts due to uterine contractions [153,154]. In this context, the presence of viral antigens from the decidual, trophoblasts, endothelial, and Hofbauer cells has been reported [67,155] indicating that the virus may initially reach the decidua through maternal blood, spread through chorionic villi, and cross the placental barrier into the fetal blood. It is known that during the process of infection in the placenta, the virus is capable of inducing the release of proinflammatory molecules such as TNF-α and INF-γ and the infiltration of TCD8+ lymphocytes. The presence of TNF-α and the infiltration of immune system cells cause placental structure alterations. This is due to the production of proteases, metalloproteinases from the granules released by these cells, cytokines, and chemokines such as MCP-1 that can trigger a series of cellular mechanisms that can culminate in cell death, which allow the virus to spread more easily to the placental tissue [16,156]. 

### 5.4. RSV

To date, knowledge on how RSV infections spread in placental tissues remains limited. However, in vitro studies have been reported that demonstrated the capacity of this virus to infect different cellular populations, including placental resident cells such as cytotrophoblasts and fibroblasts, using placental resident macrophages as reservoirs, inducing a proinflammatory environment by detecting the release of proinflammatory molecules such as IL-6, TNF-α, and INF-α [116]. In contrast, in vitro tests using trophoblast cell models, such as the BeWo cell line, are permissive to RSV infection. An important factor is the abundant expression of nucleolin, the main receptor of this virus, demonstrating the susceptibility of placental trophoblasts once the virus enters the placenta [157]. It has also been shown that human umbilical cord endothelial cells (HUVECs) are permissive to RSV infection because they present an increase in the expression of ICAM-1 (CD54) and VCAM-1 (CD106) due to the infection itself, suggesting that these adhesion molecules contribute to the accumulation of polymorphonuclear cells in the vasculature, aggravating placental damage due to inflammation [158].

### 5.5. EBOV and Other Hemorrhagic Fever Viruses

Various viruses are disseminated through blood circulation. The placenta, as a highly vascularized organ, is susceptible to becoming a target of these pathogens. However, for these viruses to reproduce in this tissue, resident cells must be permissive to infection by these viruses [122]. Therefore, in cases of infection by viruses such as EBOV, the human intracellular cholesterol transporter 1 (NPC1), which is present mainly in syncytiotrophoblasts in the placenta, is crucial for this virus to enter the cell and cause an infection. In addition, it is known that Filovirus infection can promote necrosis in infected cells, the mechanism of which can affect structures composed of trophoblasts such as chorionic villi [107,159]. It has also been reported that the virus has an ability to evade the immune response to spread in the placental tissue, inhibiting the production and response of type I interferons, which are some of the main molecules involved in the antiviral response in the placental tissue [48]. There are reports of the presence of viruses such as EBOV, LASV, and RVFV that can mainly invade syncytiotrophoblast and cytotrophoblast layers, thus inducing a state of inflammation that could explain the premature rupture of membranes or the origin of intervillitis [107,129,130]. In this context, it has been reported that EBOV and Marburg viral infections are capable of activating the expression of proinflammatory cytokines, including TNF-α, IL-6, and IL-8, in immune system cells, such as macrophages. Furthermore, these viruses aid in the production of viral glycoproteins (GPs) that can be found in circulating blood and are capable of producing a cytopathic effect and increasing the permeability of endothelial cells, thus affecting the function of the placenta. In this way, this mechanism could influence the production of hemorrhages in the placenta [66,131,132].

### 5.6. Human Coronaviruses

The tropism of this respiratory virus toward placental tissue is influenced by the presence of the main receptor that this virus uses to enter the cell. Angiotensin-converting enzyme 2 (ACE2) and transmembrane serine protease 2 (TMPRSS2) are highly expressed in stromal cells, the perivascular cells of the decidua, cytotrophoblasts, and syncytiotrophoblasts [160]. Infection with the SARS-CoV-2 virus in recent years has been reported to be capable of promoting an increase in the expression of proinflammatory mediators, resulting in exacerbated inflammation in chorionic villi which, in turn, promotes maternal and fetal vascular malperfusion [72,76]. It has been observed that infection with this virus can induce necrosis in the cells of the placental tissue, as well as high levels of infiltration of cells of the immune system so that the alteration in tissue architecture is evident; this phenomenon has been associated mainly with cases of severe infections that cause fetal death [161]. Other studies conducted in murine models have reported that infection with this virus can induce an increase in the expression of progesterone, a hormone with immunosuppressive characteristics that is involved in the antiviral response and the stimulation of progesterone receptor (PGR) by active progesterone. The tyrosine kinase SRC phosphorylates the transcription factor IRF3 at residue Y107, leading to the activation and induction of antiviral genes. Therefore, an increase in the presence of this hormone could be related to the ability of this respiratory virus to replicate successfully in the placental tissue [162].

The presence and response to type I and III interferons are crucial to responding to viral infections. It has been reported that SARS-CoV-2 infection decreases the response to interferons, perhaps as a strategy to evade the immune system [163], while it has been demonstrated in cellular models of trophoblasts, endometrium and endothelium HTR8, JEG3, PMVECs, and HEECs that when treated with the recombinant protein spike (S), it is sufficient to induce the expression of high levels of the pro-inflammatory molecules IL-6, IL-8, IL-1β, CCL-2, and CCL5 [66]. The presence of this cytokine storm has been associated with fetal death, abortions, and fetal growth restriction. In this context, it has been shown that the presence of IL-1β and IL-6 is responsible for preterm birth associated with chorioamnionitis and weakens the fetal membrane by generating metalloproteases that degrade the extracellular matrix [164,165]. Furthermore, IL-1β is a potent inhibitor of decidual cell progesterone receptor expression, leading to chorioamnionitis [165]. Therefore, the presence of these proinflammatory molecules during SARS-CoV-2 infection in pregnancy provides a basis for the origin of the aforementioned histopathological aversions.

On the other hand, it is known that the function of the mitochondria is essential to maintaining cellular homeostasis and producing adequate energy to maintain stable cellular functions and thus maintain the functionality of the placenta as a barrier and the transport of nutrients and oxygen [166,167]. In this context, it was reported that SARS-CoV-2 infection promotes alterations in mitochondrial function and oxidative stress, the latter being the result of an imbalance between reactive oxygen species (ROS) produced by mitochondria and antioxidant defenses [168,169]. This phenomenon is caused by an increase in inflammation and can similarly induce the production of a highly inflammatory environment which can induce damage in the biogenesis of the same mitochondria, affecting respiratory activity and nutrition. It has also been reported that symptomatic pregnant women present a decrease in the expression of genes that encode subunits of the respiratory chain ((NDUFA9, SDHA, and COX4I1) and genes related to mitochondrial dynamics (DNM1L and FIS1). This could explain the presence of vascular malperfusion, tissue oxygen saturation alterations, and intrauterine growth restriction [170].

Moreover, pregnant women infected with SARS-CoV-2 with severe symptoms present an increase in the expression of the Von Willebrand factor (vWf) in the vascular endothelium of the placenta, which is related to the activity of the cascade. Coagulation, an overexpression of vWf, is related to the generation of thrombosis [171] and a decrease in the expression of vascular endothelial (VE) cadherin and claudin-5, which are associated with the cell–cell junction that helps maintain endothelial barrier function. Decreased levels of expression of these proteins are associated with vascular abnormalities [52,172]. Likewise, it has been reported that endothelial cells and syncytiotrophoblasts infected with this virus can increase the expression of VEGF, whose overexpression is related to the formation of syncytial knots, causing tissue hypoxia [73]. In other studies, data have been revealed indicating a change in the presence of sialic acid, related to the morphofunctionality of placental cells [173,174], that leads to different complications such as intrauterine growth retardation, hypertensive disorders, altered glycemia, and gestational trophoblastic tumor. This is the reason for the low presence of α2,3 Galactose-linked Sias in the trophoblast observed in infected placentas, which could be associated with histopathological damage to the placenta [175]. 

## 6. Alternative Strategies as Possible Therapeutic Targets Can Regulate the Mechanisms of Placental Damage

As described previously, the main impact of viral mechanisms during an infection in pregnancy are histopathological alterations that occur in the placental tissue. Therefore, using alternative non-pharmaceutical strategies as therapeutic targets that can regulate this damage without affecting pregnant women is necessary as the consumption of drugs during pregnancy is limited. In this section, we focus on novel studies that can regulate the mechanisms of placental damage. For example, the use of extracellular vesicles such as exosomes with anti-inflammatory factors; the use of miRNAs that can regulate the expression of genes related to proliferation, differentiation, and cellular metabolism; and the application of antimicrobial peptides with immunomodulatory effects, such as Trappina-2/Elafina, among other tools could reverse or prevent placental tissue damage (Figure 4).

Because the inflammatory environment is part of the etiology of pathological changes caused by viruses, the use of exosomes as markers or carrier molecules could help develop innovative therapeutic targets to prevent the occurrence of pathological changes caused by viruses by exacerbating inflammation in the placental tissue. For example, changes in the presence of inflammatory markers such as IL-2, TNF-α, and IL-10 and other markers such as alarmin high-mobility group box 1 (HMGB1) and cell-free fetal telomere fragments (cffTFs) present in exosomes in circulation or in uterine washings during pregnancy have been proposed as indicators of the health status of the placental tissue during prenatal damage [176,177]. Recently, the use of exosomes to control inflammation and placental damage was evaluated. An in vivo study using a murine model revealed that treatment with exosomes loaded with recombinant IL-10 during pregnancy decreased inflammation caused by infections, reducing the risk of placental damage, including the effects of this phenomenon such as pre-term birth [178], a phenomenon caused by infection with some viruses, as addressed in the previous section. 

On the other hand, treatments during poor vascular perfusion of the placenta have been proposed; for example, it was reported in a pre-eclampsia rat model that silencing the protease-activated receptor 1 (PAR-1) gene promotes angiogenesis, showing histopathological results in the placenta improving microvasculature [179]. This procedure could be studied further in the context of viral infections that promote MVM in placentas.

It has also been noted that treatment with elements such as miRNAs can activate and polarize regulatory T lymphocytes to control inflammation and maintain placental homeostasis, as well as miR16, miR-Let7-c, miR-181a, miR-125b, miR-26a, miR-145, miR181c-5p, miR-Let-7e, miR-Let7-c, miR-Let-7f, and miR-106a, with great immunoregulatory potential [180,181]. In vivo and in vitro studies have also reported that the overexpression of lncRNA small nucleolar RNA host gene 5 (SNHG5) in trophoblasts and mice reverses the tissue phenotype of preeclampsia in the placenta and consequently reduces tissue damage. The overexpression of this lncRNA is associated with trophoblast proliferation, migration, and invasion and the differentiation of trophoblasts [182].

Other elements that have been tested as immunoregulators in proinflammatory conditions are small peptides such as SLPI and Trappin-2/Elafin with WAP domains, which have been proven in vitro and in vivo to inhibit and revert tissue damage caused by immune cell infiltration and the presence of cytokines and chemokines in the myocardium or intestinal epithelium affected by chronic diseases [183] or from viral infections such as the encephalomyocarditis virus or Herpes virus [184,185]. In both cases, these peptides act as immunoregulatory elements in infection and non-infection contexts [186,187]. The focus is on these peptides because they have the ability to quickly enter their target cell through diffusion, and once inside the cytoplasm, they are capable of inhibiting the activation of the transcription factor NF-kB [188], which orchestrates the activation and expression of proinflammatory genes such as IL-6, IL-8, and RANTES. We propose these peptides as alternative or complementary treatments for proinflammatory conditions caused by viral infections during pregnancy.

With this, we aim to highlight the importance of collateral damage to placental tissue during viral infections and the potential for current alternative treatments that could be adapted to revert or inhibit placental damage, mainly in a proinflammatory environment, to reduce risks to birth, fetal development, and mothers.

## 7. Conclusions

Viral infections during pregnancy remain a perinatal health problem. However, most studies have focused on how transplacental infections occur due to these pathogens that subsequently infect the fetus. Limited attention has been devoted to evaluating placental viral infections that are not transmitted vertically but negatively affect the physiology of the placenta, which alters the outcome of pregnancy. Although pathological placental damage caused by viruses such as HEV, DENV, CHIKV, RSV, EBOLA, and SARS-CoV-2, among others, has been documented, the cellular and molecular mechanisms generated by these pathogens to promote this phenomenon remain poorly understood. However, as illustrated above, placental damage effects, including fibrin deposits, placental calcification, intervillositis, chorioamnionitis, syncytial knots, and MVM during pregnancy caused by viral infection such as SARS-CoV-2, are orchestrated via the production of cytokines and the presence of a highly inflammatory environment due to the presence of viral components that white blood cells and immune system cells are capable of recognizing. However, since this is uncontrolled, it triggers a series of histological pathologies that put the health of the mother and fetus at risk. Therefore, it is essential to understand how these mechanisms occur collaterally with viral infection since treatments directed towards these mechanisms could be of great relevance for possible targeted therapeutics and improvements in delivery and product predictions.

## Figures and Tables

**Figure 1 diseases-12-00059-f001:**
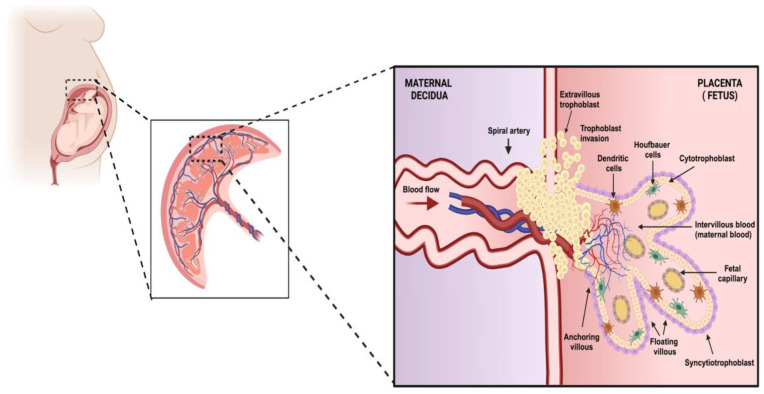
The structure and organization of placental tissue. In this image, we represent the organization of the fundamental unit of the placenta. This structure is formed by trophoblasts which are classified into cytotrophoblasts which, when differentiated toward the upper layer, give rise to the outermost membrane with multinucleated cells called syncytiotrophoblasts. In turn, the trophoblasts can differentiate into invasive extravillous trophoblasts which can reach the maternal decidua. All of these cells are part of the functional units of the placenta called chorionic villi, which can be classified according to their function into anchoring and floating villi. The primary functions of these villi are to maintain the placenta’s stability and provide gas exchange and excrete substances, respectively. Within the lumina of these structures and the decidua, we can find populations of immune system cells. This entire cellular organization is part of the fetus’s protective barrier against pathogenic agents and helps exchange substances essential for fetal development (created using BioRender.com) (Accessed on 3 March 2024).

**Figure 2 diseases-12-00059-f002:**
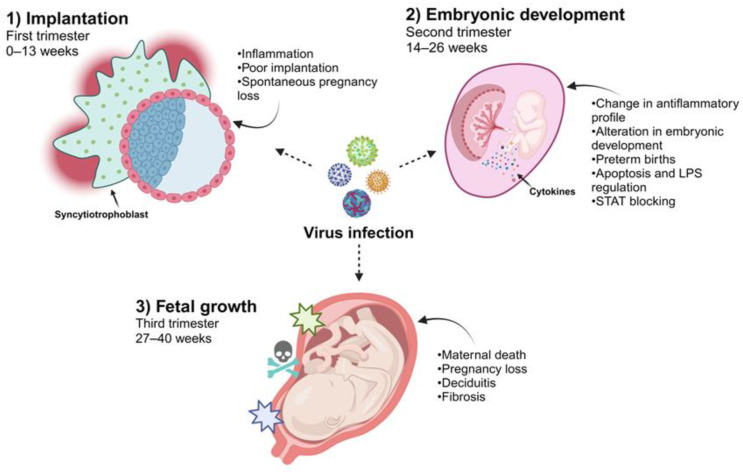
Viral infections during different stages of pregnancy. Due to the different immunological and physiological changes in the development of the placenta, some viruses may be able to infect the cells in this tissue. Viral infections during the first trimester are associated with susceptibility to EVT, affecting implantation and pregnancy loss. When infections occur during the second trimester, the main consequence is reflected in embryonic development since an alteration in inflammation that culminates in cell death, causing preterm birth, has been reported. During the third trimester, when viral infections occur, they can cause placental tissue alterations, such as fibrosis and inflammation in the decidua, resulting in the death of both the fetus and the mother (created using BioRender.com) (Accessed on 3 March 2024).

**Figure 3 diseases-12-00059-f003:**
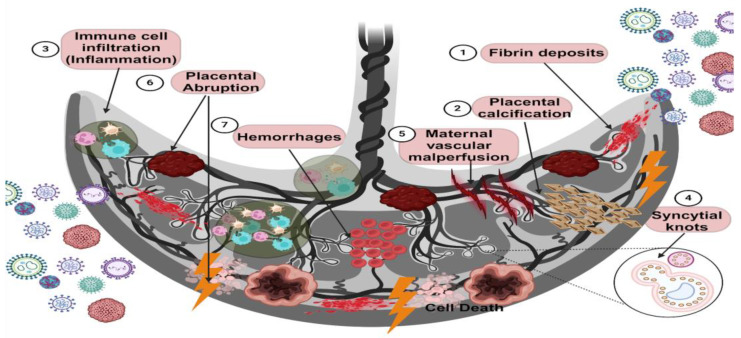
Placental damage caused by viral infections. Viral infections during pregnancy can be productive since the mother is susceptible to them. In addition, viruses can invade the placental tissue, and vertical transmission to the fetus does not occur in all cases. However, several effects of the infection on the placenta have been characterized. For example, histological alterations in the placenta during viral infections can put fetal health and development at risk. The main results of this placental damage are fibrin deposits, which are related to a coagulation disorder; placental calcification characterized by calcium crystal deposits; intervillositis and chorioamnionitis, which are characterized by the infiltration of immune system cells in both the chorionic villi and the chorion or both; syncytial knots that are represented by deposits of syncytial nuclei products of the cell death of syncytiotrophoblasts; poor maternal vascular perfusion, characterized by an alteration in the blood supply to the placental tissue, affecting the transport of nutrients and oxygen; placental abruption, characterized by detachment of the placenta into the myometrium due to the premature rupture of membranes; and hemorrhages, which are indicated by an alteration in the vascular permeability of endothelial cells that cause circulating blood to leak into the tissue. The leading collateral cause of viral infection and these aversions is an alteration in the inflammatory environment (created using BioRender.com) (accessed on 3 March 2024).

**Figure 4 diseases-12-00059-f004:**
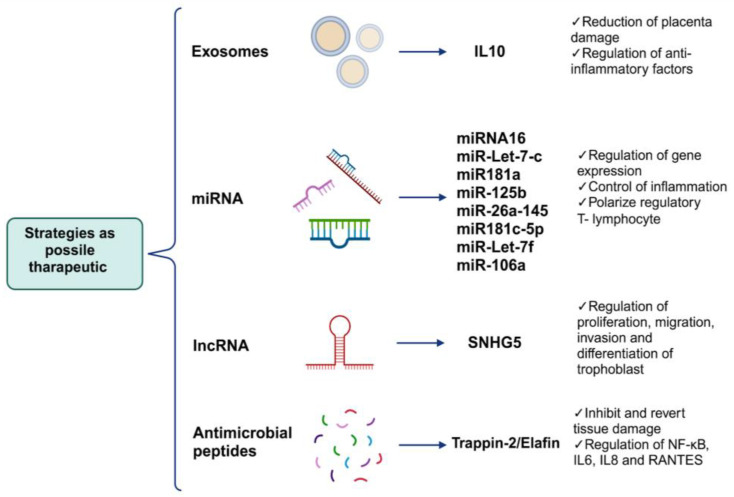
Possible alternative treatments to avoid or reverse placental damage caused by viral infections. Because placental damages collateral to viral infections are factors that put fetal survival and development at risk, below, we propose some strategies which focus mainly on some of the causes of damage as well as inflammation, which orchestrates the formation of most of the aforementioned aversions in tissue, as prophylactic treatments or for the reversal of placental damage (created using BioRender.com) (accessed on 3 March 2024).

**Table 1 diseases-12-00059-t001:** Effects on the placenta due to infection with some viruses that are not conventionally transmitted vertically but, during pregnancy, collateral effects on the placental tissue and therefore on embryonic development are reported in different stages of pregnancy.

Virus	Trimester	Effects on Placenta	References
**CHIKV**	3	Induces the release of proinflammatory molecules and alteration in the placental structure, absence of the syncytiotrophoblast layer, interruption in the placental surface, fibrosis in the stromal region, calcification, mitochondrial and endoplasmic reticulum alterations in trophoblasts, subtrophoblastic edema and cell death.	[16]
**HCMV**	1,2,3	In trophoblasts, modifies the proliferative, migratory, and invasive properties necessary for this implantation.	[21,22,23]
**HSV**	1,2,3	Disturbs the transport of antigens via HLA-G, affecting the tolerance of immune cells and thereby generating mechanisms of damage to the maternal–fetal interface.	[27,28]
**HPV**	1	Promotes an apoptotic effect, implantation, early embryonic development and miscarriage.	[31,32,33,34]
**LASV**	3	Regulation of interferon response.	[40]
**SARS-CoV-2**	1	Poor maternal vascular malperfusion, increased fibrin deposition, villous agglutination, intervillous thrombi, atherosis and increased syncytial knots, calcification, manifest placental damage associated with inflammation, manifesting intervillitis, with high infiltration of neutrophils and Hofbauer cells in the chorionic villi, as well as cell death.	[50,51,52,53]
**HEV**	3	Calcification, fibrosis, and great tissue inflammation, membrane ruptures, hemorrhages, expression of proinflammatory cytokines and high infiltration of immune system cells.	[54,55,56,57]
**DENV**	1	Induces the release of proinflammatory molecules, inhibit the activation of the cGAS/STING pathway, decrease the activation of the mTOR pathway related to cellular processes such as growth and differentiation.	[58,59]
**RSV**	3	Alterations in the membranes of syncytiotrophoblasts and placental damage due to inflammation.	[60,61,62]
**HBoV**	1,2,3	Spontaneous abortions.	[63]
**EBOV**	1,2,3	Expression of proinflammatory cytokines, permeability of endothelial cells and hemorrhages in the placenta.	[64,65]

**Table 2 diseases-12-00059-t002:** The main adverse effects on placental tissue indicated in the figure above, caused by the viruses listed in this table, are described.

Changes in the Placenta	Virus	References
1. Fibrin deposits	Adenovirus, Coxsackievirus–B, SARS-CoV-2	[51,52,66]
2. Placental calcification	HEV, CHIKV, SARS-CoV-2	[58,67,68]
3. Placental inflammation (Intervillositis and Chorioamnionitis)	HEV, DENV, CHIKV, SARS-CoV-1, SARS-CoV-2, RSV, EBOV	[16,52,59,66,69,70,71,72]
4. Syncytial knots	RSV, SARS-CoV-2	[53,73,74]
5. Maternal vascular malperfusion	SARS-CoV1, SARS-CoV-2	[72,75,76]
6. Placental abruption	HBV, SARS-CoV-2	[77,78]
7. Hemorrhages	HEV, DENV, EBOV, MARV, LASV, RVF	[58,70,79,80]

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
