# Peer review of "Collateral Damage in the Placenta during Viral Infection in Pregnancy: A Possible Mechanism for Vertical Transmission and an Adverse Pregnancy Outcome"

_diseases, 2024, doi:10.3390/diseases12030059_

Round 1
Reviewer 1 Report
Comments and Suggestions for Authors
This is an interesting article in which the authors reviewed different viral infections and outcomes in placental cells that could affect the mother and/or the fetus. In addition, the authors revised the likelihood that some viruses are transmitted from mother to fetus (vertical transmission), as well as the mechanisms involved.
In my opinion the manuscript is well organized and explained, I only have some minor comments and suggestions aimed at improving the final version.
Abstract:
The abstract starts with this sentence: “A placenta is a connection between a mother and a new developing organism”. I suggest a small change to be more specific: “In humans (or in mammals) the placenta is a connection…”
Introduction:
This section could use a figure; lines 49 to 69 describe the placental structure and the different cell types, this could be illustrated schematically with a figure.
Some paragraphs are too long; the authors can probably break up some paragraphs to make reading easier.
Figure 1:
Revise the figure, it has typos:
1. FIRST trimester
2. Preterm BIRTHS
3. Third TRIMESTER; Maternal death (instead of Death maternal)
I suggest that the Table below the figure be presented as a separate table (Table 1), and not as part of the figure. Please add a Reference column.
Hepatitis E Virus:
Lines 291-292 mention that “…pregnant women were infected with HEV, of which seven died in the first week due to premature birth…” I think this phrase needs clarification, since “premature birth” refers to babies born before 37 weeks of gestation, in extreme cases, less than 28 weeks, however at the end of the first week the embryo is an implantation blastocyst, I wonder if it is correct to say premature birth at this early stage, and if this premature birth would really result in death of the mother.
Chikungunya Virus:
Lines 332-335: “…it has been reported in some meta-analyses that approximately 50% of infected pregnant women can vertically transmit the virus; however, in most cases, it is intrapartum at the time of delivery (75 by cesarean section and 96 natural deliveries)…” the authors meant 75% and 96% respectively, right? If this is correct please include the percentage (%) sign to avoid any confusion.
Figure 2:
Revise the use of periods and semicolons in the figure legend.
Figure 3:
miRNAs - REGULATION OF GENE EXPRESSION
Conclusion:
This phrase is not very clear: “…understanding these mechanisms produced collaterally with the viral infection is essential…” I suggest a small change: “…understanding how these mechanisms are produced collaterally with the viral infection is essential…” or “… it is essential to understand how these mechanisms occur collaterally with viral infection…”
Comments on the Quality of English Language
Minor editing required.
Author Response
Dear Reviewers
I am pleased to resubmit for publication the revised version of “Collateral damage in the placenta during viral infection in pregnancy: A possible mechanism for vertical transmission and adverse pregnancy outcome”. I appreciated the constructive criticism from the associated editor and reviewers. I have addressed each of their concerns as outlined below.
Following the reviewer’s advice, I, along with my collaborators have been carefully revised and appropriate changes have been made in accordance with the reviewer’s suggestions. The responses to their comments are provided below:
Reviewer 1
1.- Abstract: The abstract starts with this sentence: “A placenta is a connection between a mother and a new developing organism”. I suggest a small change to be more specific: “In humans (or in mammals) the placenta is a connection…”
We appreciate this comment; we have made changes to the text.
2.- Introduction: This section could use a figure; lines 49 to 69 describe the placental structure and the different cell types, this could be illustrated schematically with a figure.
We appreciate your valuable comment. We designed a figure that shows this cellular structure and thus reinforces the information that is in the text.
3.- Some paragraphs are too long; the authors can probably break up some paragraphs to make reading easier.
We appreciate your observation, and we have fragmented some paragraphs to give a better reading of the text.
4.- Figure 1:
Revise the figure, it has typos:
- FIRST trimester
- Preterm BIRTHS
- Third TRIMESTER; Maternal death (instead of Death maternal)
I suggest that the Table below the figure be presented as a separate table (Table 1), and not as part of the figure. Please add a Reference column.
Thank you very much for your observation; we have modified the recommendations.
5.- Hepatitis E Virus:
Lines 291-292 mention that “…pregnant women were infected with HEV, of which seven died in the first week due to premature birth…” I think this phrase needs clarification, since “premature birth” refers to babies born before 37 weeks of gestation, in extreme cases, less than 28 weeks, however at the end of the first week the embryo is an implantation blastocyst, I wonder if it is correct to say premature birth at this early stage, and if this premature birth would really result in death of the mother.
We consider your comment relevant, and we have modified the requested information.
6.- Chikungunya Virus:
Lines 332-335: “…it has been reported in some meta-analyses that approximately 50% of infected pregnant women can vertically transmit the virus; however, in most cases, it is intrapartum at the time of delivery (75 by cesarean section and 96 natural deliveries)…” the authors meant 75% and 96% respectively, right? If this is correct please include the percentage (%) sign to avoid any confusion.
We appreciate your comments and observations, which were considered; we have modified the information in the text.
7.- Figure 3:
miRNAs - REGULATION OF GENE EXPRESSION
We appreciate your comments, and we have modified the requested information.
8.- Conclusion:
This phrase is not very clear: “…understanding these mechanisms produced collaterally with the viral infection is essential…” I suggest a small change: “…understanding how these mechanisms are produced collaterally with the viral infection is essential…” or “… it is essential to understand how these mechanisms occur collaterally with viral infection…”
We consider your comment relevant, and we have modified the requested information.
I look forward to your reply.
Sincerely,
Dr. Moisés León Juárez
Investigador en ciencias médicas D
Laboratorio de Virología Perinatal y
Diseño Molecular de Antígenos y Biomarcadores
Departamento de Inmunobioquimica
Instituto Nacional de Perinatología
moisesleoninper@gmail.com
Tel 55209900 ext222
Reviewer 2 Report
Comments and Suggestions for Authors
This is a reasonably good research paper but there are some shortcomings that need to be addressed:
1) Minor: A table is incorporated into Figures 1 and 2. The Tables should be placed as Table 1 and 2 and you can refer to the Tables in the legends of the figures.
2) I am a bit puzzled about the emphasis on SARS-CoV-2, EBOV and DENV on the placenta. All viruses are likely to have some effects on the placenta. But unlike Zika and HIV, these viruses are not particularly known to have the most direct impact on the fetes:
https://www.mayoclinic.org/diseases-conditions/coronavirus/in-depth/pregnancy-and-covid-19/art-20482639
Even it is not unusual for most viruses including those emphasised by the authors, Zika and HIV are known for their particular danger to pregnant women and their foetuses.
https://www.ncbi.nlm.nih.gov/pmc/articles/PMC7490298/
https://www.nature.com/articles/s41467-017-02499-9
https://www.ncbi.nlm.nih.gov/pmc/articles/PMC7192492/
Given the more serious risks of HIV and Zika, why isn't greater emphasis given to HIV and Zika?
3) There are important reasons, that Zika and HIV are a greater threat to the placenta. Computational models have shown that HIV, Zika and Yellow Fever virus have more disordered outer shell and as a result the viruses are able to enter organs more easily as greater disorder in a protein enables for efficient binding.
https://pubmed.ncbi.nlm.nih.gov/31072073/
https://pubmed.ncbi.nlm.nih.gov/31698857/
DENV is a closely related to Zika but while DENV has been of medical interest for a long time, unlike Zika. DENV has been known to cause severe diseases to a small percentage of infected people but its impact on the placenta is nothing like Zika. Zika on the other hand, has been known since 1947 but it was not medically interesting as the symptoms are much milder than DENV. It became interesting only during the huge outbreak in Americas in 2015-16 , when a large numbers of babies were born with microcephaly as a result of the breaching of placenta. The computational model above showed that DENV has a rigid outer shell and disordered inner shell but Zika has a disordered outer shell and rigid inner shell. Yellow Fever Virus has both highly disordered inner and outer shells. YFV is among the most virulent virus known. HIV also has a disordered outer shell. What are the proteins in the placenta that the outer shell can easily bind to such that it allows easy penetration of HIV and Zika of the placenta?
Comments on the Quality of English Language
The sentence on Line 51-53 is not grammatically correct. If you want to put a ";" on line 51. the phrase after ";" has to be a complete sentence. You may be able to get away if you replace ";" with a ":" on line 51
Author Response
Dear Reviewers
I am pleased to resubmit for publication the revised version of “Collateral damage in the placenta during viral infection in pregnancy: A possible mechanism for vertical transmission and adverse pregnancy outcome”. I appreciated the constructive criticism from the associated editor and reviewers. I have addressed each of their concerns as outlined below.
Following the reviewer’s advice, I, along with my collaborators have been carefully revised and appropriate changes have been made in accordance with the reviewer’s suggestions. The responses to their comments are provided below:
Reviewer 2
1.- This is a reasonably good research paper but there are some shortcomings that need to be addressed:
1) Minor: A table is incorporated into Figures 1 and 2. The Tables should be placed as Table 1 and 2 and you can refer to the Tables in the legends of the figures.
2) I am a bit puzzled about the emphasis on SARS-CoV-2, EBOV and DENV on the placenta. All viruses are likely to have some effects on the placenta. But unlike Zika and HIV, these viruses are not particularly known to have the most direct impact on the fetes:
https://www.mayoclinic.org/diseases-conditions/coronavirus/in-depth/pregnancy-and-covid-19/art-20482639
Even it is not unusual for most viruses including those emphasised by the authors, Zika and HIV are known for their particular danger to pregnant women and their foetuses.
https://www.ncbi.nlm.nih.gov/pmc/articles/PMC7490298/
https://www.nature.com/articles/s41467-017-02499-9
https://www.ncbi.nlm.nih.gov/pmc/articles/PMC7192492/
Given the more serious risks of HIV and Zika, why isn't greater emphasis given to HIV and Zika?
3) There are important reasons, that Zika and HIV are a greater threat to the placenta. Computational models have shown that HIV, Zika and Yellow Fever virus have more disordered outer shell and as a result the viruses are able to enter organs more easily as greater disorder in a protein enables for efficient binding.
https://pubmed.ncbi.nlm.nih.gov/31072073/
https://pubmed.ncbi.nlm.nih.gov/31698857/
DENV is a closely related to Zika but while DENV has been of medical interest for a long time, unlike Zika. DENV has been known to cause severe diseases to a small percentage of infected people but its impact on the placenta is nothing like Zika. Zika on the other hand, has been known since 1947 but it was not medically interesting as the symptoms are much milder than DENV. It became interesting only during the huge outbreak in Americas in 2015-16 , when a large numbers of babies were born with microcephaly as a result of the breaching of placenta. The computational model above showed that DENV has a rigid outer shell and disordered inner shell but Zika has a disordered outer shell and rigid inner shell. Yellow Fever Virus has both highly disordered inner and outer shells. YFV is among the most virulent virus known. HIV also has a disordered outer shell. What are the proteins in the placenta that the outer shell can easily bind to such that it allows easy penetration of HIV and Zika of the placenta?
We appreciate the reviewer's comments, and we agree that the effect that viruses such as Zika and HIV-1 generate in the context of pregnancy has been widely documented since their vertical transmission is associated with the development of problems in the future health of the fetus. The permissiveness and susceptibility of placental components towards these viruses is one of the issues that may be associated with promoting the placenta to function as a bridge to transfer the infection to the fetal circulation. In such a way, various reviews have brought together this evidence and clarified that these pathogens have relevance in perinatal medicine. However, our review is trying to address another situation; as the reviewer rightly mentions, viruses can promote damage or affect the placenta. However, little has been documented as viruses that do not develop a vertical infection because they do not promote an infection. Directly on the fetus, they are generating a system of damage due to pathogens in the mother. We describe these effects since the viral infection compromises the physiology of the placenta and brings with it perinatal complications that can disturb maternal and fetal health.
For this reason, we emphasize that respiratory or vector-transmitted infections should not be considered a situation that cannot affect the proper development of the pregnancy. For this reason, we believe that our manuscript gives a different approach to other reviews that deal with the topic, considering viruses that, although their vertical transmission is limited or null, the examples we address in the text help us understand that this topic could be relevant in perinatal medicine.
Finally, we again thank you for your suggestions and insights, which have enriched the manuscript and produced a more balanced and better account of the review. We hope that the revised manuscript is now suitable for publication in the prestigious journal that you represent.
I look forward to your reply.
Sincerely,
Dr. Moisés León Juárez
Investigador en ciencias médicas D
Laboratorio de Virología Perinatal y
Diseño Molecular de Antígenos y Biomarcadores
Departamento de Inmunobioquimica
Instituto Nacional de Perinatología
moisesleoninper@gmail.com
Tel 55209900 ext222
Round 2
Reviewer 2 Report
Comments and Suggestions for Authors
Improvements seen